# Investigation of the Exercise Dependence of Athletes Doing Kickboxing, Taekwondo, and Muay Thai

**DOI:** 10.3390/sports7020052

**Published:** 2019-02-25

**Authors:** Serdar Orhan, Ali Serdar Yücel, Bootan Jawhar Sadeq, Ebru Orhan

**Affiliations:** 1Faculty of Sports Science, Firat University, Elazig 23119, Turkey; alsetu_23@hotmail.com; 2High School of Erbil, Erbil 44001, Iraq; botan_basket@yahoo.com; 3Institute of Health Sciences, Physical Education and Sports, Firat University, Elazig 23119, Turkey; ebruorhan023@gmail.com

**Keywords:** exercise dependence, symptomatic, asymptomatic, Kickboxing, Taekwondo, Muay Thai

## Abstract

Debates about the conditions in which the frequency and intensity principles of regular exercise, depending on the fact that a sports background can be accepted as extremism, are still a controversial topic. The purpose of this research was to investigate the exercise dependence of athletes who practice Kickboxing, Taekwondo, and Muay Thai. The study included 141 athletes, consisting of 87 men and 54 women. The Exercise Dependence Scale-21 (EDS-21), composed of 21 items developed by Hausenblas and Downs and adapted into the Turkish version by Yeltepe and İkizler, was applied to the athletes. As a result of the research, while athletes showed more sensitivity to the EDS (=71.41), this scale was also defined as symptomatic. It was found that five athletes (3.5%) were asymptomatic-nondependent, 117 athletes (83.0%) were symptomatic-nondependent, and 19 athletes (13.5%) were at risk for exercise dependence. It was determined that athletes were at risk for exercise dependence as follows: Eight athletes were doing Kickboxing, ten athletes were doing Taekwondo, and one athlete was doing Muay Thai. A significant difference was observed according to years of regular training and number of trainings per a day. Other variables presented no significant differences. It was possible to say that years of regular training could be effective in revealing exercise dependence.

## 1. Introduction

Exercise helps to prevent and improve a number of health problems, including high blood pressure, diabetes, depression, anxiety, premenstrual syndrome, and stress. The physical benefits of exercise also include reducing risk of coronary heart disease, ensuring weight control, flexibility, improving muscle strength and durability, and reducing back and waist problems [1,2,3,4].

Regular physical activity has a number of effects on mental health, such as reducing depression and anxiety, improving sleep, providing relief, and enhancing physiological benefits with self-esteem. However, uncontrolled excessive exercise and physical activity are often defined as exercise dependence [4,5]. Nevertheless, not all people exercising with increasing frequency and intensity should be considered as dependent on exercise. Those who do not have problems with their exercise regimen can only be considered as highly engaged in this behavior [6].

Exercise dependence is defined as the exercise routine controlling the individual, the duration of the exercise with increasing frequency and severity, the inability to take time away from exercising to see family and friends, to exercise instead of participating in social activities, and rehabilitating the individual’s life within the framework of exercise habits [7,8]. In another definition, exercise addiction is defined as “an exercise that is incompatible with the desire for physical activity creates negative psychological symptoms such as physical injury, anxiety and depression” [9].

Some studies have shown that exercise dependency can change by factors, such as age, gender, educational status, exercise duration, and exercise frequency [5,10,11,12]. There are scales for the evaluation of excessive physical exercise, in addition to the studies indicating which sports predispose more to sports dependence [8]. For example, exercise dependency is observed in bodybuilders more than the other athletes [10].

Exercise dependence, in particular, expresses the desire to exercise with a strong emotion in every leisure time of a person [13]. It manifests itself with physiological symptoms (tolerance, avoidance) or psychological symptoms (depression, tension) in the exercise process. Hausenblas and Symons explain this with the term “Exercise Dependence”. The prevalence of exercise dependence is not known. However, quite a few of the male and female athletes have stated that they suffer from heavy dependency. Multidimensional measurements of exercise dependence show that men are addicted to exercise more than women [14]. In contrast, women tend to be more addicted to weight loss as compared to men [7,15].

Researchers have found that exercise dependence manifests itself negatively with anxiety, depression, nervousness, and insomnia [16] and positively when the person doing excessive exercise cannot do it and they do exercise in order to cope with the difficulties encountered by the individual in her/his life [17]. Exercise dependence is broadly stated to be associated with such factors as personality traits, psychological and physiological factors, type of exercise, gender, and years of participation in exacerbations [16].

Szabo and Griffiths (2007) indicated that the manifestation of exercise dependence symptoms and the prevalence of exercise dependence at-risk can be affected from the sample studied [18]. According to a recent national survey, the prevalence of exercise dependence at-risk ranges from 0.3% to 0.5% in the general adult population [19,20]. Higher prevalence rates were reported in other groups as follows: 3.2% in athletes competing in ultra-marathons [6], 3.6% in the population doing general exercise, 6.9% in British students studying sports science [18], 5.1% in college students [21], and 15.1% in bodybuilders, powerlifters, and fitness lifters [22]. Among the clients of a fitness center in France, the prevalence rate for exercise dependence was found as 42% [23], whereas the same prevalence was established as 45.9% among college students in another study [7]. Results obtained from other studies suggest that levels of exercise dependence and body dissatisfaction are even higher among marathon runners and body builders [15,17]. Blaydon and Lindner (2002) found that 28% of nonprofessional athletes experience exercise dependence [24]. Other studies conducted in college-aged populations exerted that 21.8% of participants displayed obligatory or dysfunctional activity patterns [12] and 18.1% reported compulsive exercise [25]. Literature data on the epidemiology of sports dependence are preliminary and inconsistent. Most probably, significant variations between different studies regarding the prevalence of sports dependence exist because of the unclear definition of dependence, aside from the use of ineffective evaluation tools.

A new behavioral addiction study stated that the problem of physical exercise has started to become a problem of extremism and has been reported to be based on different etiopathogenic theories. According to these theories, the concepts of addiction, dependence, obligatory exercise, compulsive exercise, or driven exercise become prominent [8,26]. In the current paper, the term exercise dependence was used to be consistent with the name of the survey used (i.e., the Exercise Dependence Scale-Revised (EDS-R)) and the terminology used in previous investigations that incorporated the EDS-R [21,22,27].

Although there are limited studies on exercise dependence in the Turkish population, there are no studies conducted for sports branches. The definitions for exercise dependency are not yet complete. There is a need for further studies to contribute to the literature and to focus on the status of these factors in different sports branches. The purpose of this research was to determine the differences in exercise dependence of individuals participating in Kickboxing, Taekwondo, and Muay Thai exercises, which require special discipline and regular training by the exercise type, exercise age, and loading dynamics.

## 2. Materials and Methods

The general screening model is a descriptive research method used in this research study.

### 2.1. Participants

The athletes engaged in Kickboxing, Taekwondo and Muay Thai constituted the research population and sample. The participants were 141 (*n* = 87 men and *n* = 54 women) athletes composed of 76 in Kickboxing, 28 in Taekwondo, and 37 in Muay Thai. Their age ranged from 18 to 40 years old. An a priori power analysis, conducted using G*Power (version 3) [28], ensured that the sample sizes were sufficient to yield adequate statistical power for the procedures conducted in our study. More specifically, to detect a significant finding (at the 0.03 level) at a desired power level of 0.95, a minimum of 111 participants were required.

### 2.2. Procedures 

Participants were recruited from eight clubs selected from 97 sports clubs of various sports branches active in the province of Elazig in Turkey. These eight clubs were Kickboxing clubs (four), Taekwondo clubs (two), and Muay Thai clubs (two). The athletes in all eight clubs included in the study were aged 18 and older and had at least one year of regular exercise experience. Face-to-face interview techniques were applied, and all participants were informed about the study. Participation in the study was voluntary, and the participants had the option to withdraw from the study at any time without providing any reason for their decision. After describing the nature of the study, interested individuals voluntarily completed an institutionally-approved informed consent. In a separate room of the sports hall, participants completed the questionnaire form, composed of a personal information form and exercise dependence scale within a 5-min evaluation process. Participants were asked to mark the option that most reflected their status. A research assistant was present to monitor the participants’ progress and provide assistance. Of the 198 persons who signed the informed consent, 33 were withdrawn due to work and 24 were not evaluated because their questionnaire was incorrect/incomplete. Thus, this study included survey of 141 athletes, composed of 87 males and 54 females.

### 2.3. Measures

#### 2.3.1. The Personal Information Form

The personal information form was prepared by the researcher in order to better carry out the research with a personal information form. It was prepared as a form in which questions related to self-reported gender, age, marital status, education level, sports branch, job, sports age, years of regular training, days of exercise per a week, number of daily training, daily training time, satisfaction with the physical appearance, and reason for training were included.

#### 2.3.2. Exercise Dependence Scale (EDS-21) 

The EDS, composed of 21 items developed by Hausenblas and Downs [29] and adapted into Turkish by Yeltepe and İkizler [30], was administered to athletes. The EDS-21 is a Likert type (never-1 and always-6) measure consisting of 21 questions developed to determine exercise dependence. The EDS-21 has seven sub-dimensions. These are: (1) Withdrawal (I am exercising to avoid tensions), (2) Continuity (I am exercising even when I am hurt), (3) Tolerance (I am constantly increasing exercise intensity to improve the desired effect), (4) Control Loss (I am not reducing my frequency of exercise), (5) Decrease of other activities (I think about exercise even if I have to focus on work or lessons), (6) Time (I spend a lot of time on the exercise), and (7) Effect of Intention (I exercise longer than I plan). In the calculation of the total average scores for the EDS-21, a higher score indicates more exercise dependence symptoms. In order to determine the individuals who are classified into the dependent range of 3 or more of the EDS-21, criteria are classified as exercise dependent. The dependent range is operationalized as indicating a score of 5 or 6 for that item. Individuals who scored in the range of 3 to 4 are classified as symptomatic. These individuals may theoretically be considered at risk for exercise dependence. Finally, individuals who score within the range of 1–2 are classified as asymptomatic [29]. The total score of EDS-21 is between minimum 26 and maximum 126 points. Individuals who scored in the 0 to 42 range are classified as asymptomatic, while scores in the 43 to 84 range are classified as symptomatic and scores in the 85 to 126 range are classified as dependent. A cut-off score of 85 or more identifies individuals considered at risk for exercise dependence. The internal consistency coefficient (Cronbach Alfa coefficient) of the scale was found to be 0.95 by Hausenblas and Downs [17] and to be 0.97 by Yeltepe and İkizler [30]. The internal consistency coefficient of this study was found to be 0.97.

### 2.4. Ethical Approval

All participants gave their informed consent for inclusion before they participated in the study. The study was conducted in accordance with the Declaration of Helsinki, and the protocol was approved by the Ethics Committee of Firat University Non-Interventional Research (04.05.2017-08/15).

### 2.5. Statistical Analysis

Within the scope of the research, descriptive statistics were used to summarize the demographic and personal information of the sample group, and the data were evaluated with the statistical package program for the SPSS (version 21, IBM, New York, NY, USA). In addition, the data obtained were analyzed in statistical package program. According to the normality test, the data were seen not to be normally distributed. The Mann–Whitney U test was used for comparison of two independent variables, while the Kruskal–Wallis H test for one-way ANOVA was used in the intergroup comparisons. In different groups, Dunnett’s T3 (or Dunnett’s-HSD) was used to determine from which group the difference was derived. Significance level was accepted as 0.05 and 0.01.

## 3. Results

The findings obtained in this study are expressed in the following tables. According to Table 1, 59.6% of respondents were male, 59.6% of respondents were in the 18–22 age group, 66.7% of respondents were single, 39.0% of respondents were at the high school level, 53.9% of respondents were Kickboxing, 55.3% of respondents were students, 31.9% of respondents were 1–2 years of sports age, 35.5% of respondents had 1–2 years of regular training, 43.3% of respondents exercised 3–4 days per week, 59.6% of respondents were training one session per a day, 36.2% of respondents trained 60–90 min daily, 51.1% of respondents were satisfied with their physical appearance, and 33.3% of respondents were training to be successful.

According to Table 2, when the responses of the participants to the exercise dependence scale were examined, 26.00 was found to be minimum value and 107.00 maximum value, with an average score of 71.41. This scale was defined as symptomatic with a score of 3.40.

According to Table 3, in categorizing the exercise dependence averages of the participants, it was found that five athletes (3.5%) were asymptomatic-nondependent, 117 athletes (83.0%) were symptomatic-nondependent and 19 athletes (13.5%) were at risk for exercise dependence. There was a significant difference between the symptoms of exercise dependence. The symptomatic group was significantly different from the asymptomatic group and the dependent group (F = 150.882, *p* = 0.000).

In Table 4, the percentage ratios between the variables and the exercise dependence symptoms were given as cross-comparisons. The highest values are as follows; 5.7% of asymptomatic are male, 1.2% of asymptomatic are in the 18–22 age group, 10.3% of asymptomatic are married, 6.7% of asymptomatic are at the degree level, 5.3% of asymptomatic are Kickboxing, 2.6% of asymptomatic are students, 8.9% of asymptomatic are 1–2 years of sports age, 10.0% of asymptomatic are 1–2 years of regular training, 10.7% of asymptomatic are irregular exercise per a week, 6.0% of asymptomatic are 1 training per a day, 3.9% of asymptomatic are 60–90 min training time per a daily, 7.4% of asymptomatic are partially satisfied with the physical appearance and 7.3% of asymptomatic are training to be healthy.

The highest values are as follows; 79.3% of symptomatic are male, 84.5% of symptomatic are in the 18–22 age group, 81.9% of symptomatic are single, 76.4% of symptomatic are at the high school level, 84.2% of symptomatic are Kickboxing, 82.1% of symptomatic are students, 82.2% of symptomatic are 1–2 years of sports age, 84.0% of symptomatic are 1–2 years of regular training, 80.3% of symptomatic are 3–4 days exercise per a week, 75.0% of symptomatic are 1 training per a day, 78.4% of symptomatic are 60–90 min training time per a daily, 83.3% of symptomatic are satisfied with the physical appearance and 85.4% of symptomatic are training to be successful.

The highest values are as follows; 14.9% of dependent are male, 14.3% of dependent are in the 18–22 age group, 16.0% of dependent are single, 21.8% of dependent are at the high school level, 35.7% of dependent are Taekwondo, 15.4% of dependent are students, 39.6% of dependent are 3–6 years of sports age, 25.6% of dependent are 3–4 years of regular training, 18.0% of dependent are 3–4 days exercise per a week, 19.0% of dependent are 1 training per a day, 17.6% of dependent are 60–90 min training time per a daily, 15.3% of dependent are satisfied with the physical appearance and 21.3% of dependent are training to be successful.

Also, exercise dependence scores were compared with variables. In Table 4 according to Mann-Whitney U test results, there is no significant difference between the variable of gender and exercise dependence scores (U = 2319, *p* = 0.845). According to one-way ANOVA (Kruskal–Wallis), when the averages of exercise dependence were compared with the variables, it was seen that there was a significant difference between years of regular training (x^2^ = 7.192, *p* = 0.027) and training per a day (x^2^ = 11.023, *p* = 0.004). It was determined that the participants of the study did not have a statistically significant difference between the age (x^2^ = 4.906, *p* = 0.086), marital status (x^2^ = 2.732, *p* = 0.255), education level (x^2^ = 3.249, *p* = 0.197), sports branch (x^2^ = 1.892, *p* = 0.388), job (x^2^ = 0.034, *p* = 0.983), sports age (x^2^ = 5.337, *p* = 0.069), days exercise per a week (x^2^ = 5.743, *p* = 0.057), training time per a daily (x^2^ = 0.095, *p* = 0.954), satisfied with the physical appearance (x^2^ = 1.863, *p* = 0.394) and reason for training (x^2^ = 4.262, *p* = 0.119) compared to the total exercise dependence score.

According to Table 5, post-hoc analyses were made to determine in which group the difference originated. It was found that there were significant differences between the asymptomatic group and other groups, according to years of regular training (Dunnett’s-HSD), *p* = 0.000). Significant differences were also found between the symptomatic group and other groups, according to number of training per a day (Dunnett’s-HSD, *p* = 0.000).

## 4. Discussion

A total of 141 individuals participated in this study in order to determine the factors that could affect exercise dependence in Kickboxing, Taekwondo, and Muay Thai athletes. There were 87 male and 54 female athletes, with 76 athletes in Kickboxing, 28 in Taekwondo, and 37 in Muay Thai.

Studies performed on exercise addiction are mostly conducted with university students. Therefore, knowledge on the prevalence of exercise dependence in different age groups is limited [19,20]. In this study, a large majority (32.9%) of the participants were reported to be students [11]. About half of the respondents were students in this study.

Costa reported that the number of male adults (25–44 years) reported to be at risk of exercise dependence are significantly higher than younger male adults (18–24 years), whereas no difference was ascertained in females with respect to age groups [31]. However, our results are similar to the study of Hale et al., which failed to find age difference in exercise dependence between young adults (18–24 years) and adults (25–55 years) [22]. Araz et al. (2007) performed a study with individuals between the ages of 18 and 80 and determined that young individuals prefer exercise mostly when exercise and sports are among the behaviors ensuring health [32]. In their study, Ergun and Güzel (2018) found that exercise dependence is more common among single participants [33]. This may have been due to a higher level of perceived loneliness among single people [34], the greater importance of self-respect, and a greater desire to be liked.

Relative to the dependency status, 83.0% of the participants were in the symptomatic group, who were not dependent but at risk of dependence. When categorizing the exercise dependence mean values of the participants, five athletes (3.5%) were found to be asymptomatic-nondependent, 117 athletes (83.0%) were symptomatic-nondependent, and 19 athletes (13.5%) were at risk for exercise dependence. In addition, there were significant differences between the symptomatic group and the other groups, according to the total scores of the scale. The studies conducted by Yeltepe and İkizler [30] and Zırhlıoğlu [10] also put forth that the proportion of participants in the non-dependent symptomatic group was high. Similar findings were found in the study of Bavli et al. [11]. Although the incidence of those at risk for exercise dependence was low in the study groups, it was supported with study in previous ones that symptomatic groups bear a high risk of dependence in the exercise dependency classification [5,10,11,16,35].

Exercise dependence is not a frequently observed phenomenon, as noted in previous studies. These studies show that the rate of exercise dependence is between 3% and 12% [5,11,16]. The prevalence of risky exercise dependence in this study (13.5%) appeared to be above the range of rates ascertained in other subject groups [18,21,27]. However, a recent national survey reported much lower rates in the general adult population (0.3% to 0.5%) [19], and another study similarly revealed much higher rates in the sample of bodybuilders, powerlifters, and fitness lifters (15.1%) [22]. In light of these data, Kickboxing, Taekwondo, and Muay Thai athletes may be likely to participate in exercise activities that increase muscle strength and endurance; thus, future research should discover that sports may be more likely to show signs of dependence in exercises that require patience, endurance, and strength.

It was observed in this study that the mean scores of the asymptomatic group performing regular training were significantly lower than the other groups. According to the number of daily trainings, the mean score of the symptomatic group was significantly higher than the other two groups. Looking at the studies performed in this field, it was determined that high frequency exercise and year of exercise are closely related to exercise dependence [11,17]. In addition, also supported by Başoğlu‘s study, the increase in the number of days and the number of hours of exercise seems to be a premise of exercise dependence [36]. Based on these findings, it was possible to say that the risk of exercise dependence was high in individuals who exercise regularly. It was also determined that the participants of the study did not have a statistically significant difference between gender, age, marital status, education level, sports branch, job, exercise age, exercise days per a week, the daily training time, satisfied with the physical appearance, and reason for training, compared to the total exercise dependence score.

In the study, 13 men and 6 women were determined to be at risk for exercise dependence, while 69 men and 49 women were stated as symptomatic non-dependent. In the study, no statistically significant difference existed between the genders. Berczik et al. (2012) reported that men reveal exercise dependence symptoms more than women [26]. Costa found that there are no significant gender differences in the number of participants classified as at-risk for exercise dependence [31]. Lichtenstein et al. (2014) established that exercise dependence is more common among men [37]. Polat and Şimşek (2015) conducted a study in Eskisehir, Turkey, and ascertained that men do exercise more, the rate of exercise dependence is low for both genders, and no statistically significant difference exists between the genders [16]. Our results were similar to those of previous research.

Of the 19 athletes who were at risk for exercise dependence, eight athletes did Kickboxing, ten athletes did Taekwondo, and one athlete did Muay Thai. It was determined that 117 athletes exhibited the characteristics of being symptomatic non-dependent, of which 64 athletes did Kickboxing, 17 athletes did Taekwondo, and 36 athletes did Muay Thai. There was no statistically significant difference between the sports branches in the study. Since no other studies regarding this field existed in the literature, it could not be compared. But, similar studies determined that the participants preferred exercising regularly in football (38.3%) [10], and they preferred exercises like running, jogging, and weight lifting in sports halls [11]. Yeltepe and Ikizler established that athletes at risk of exercise dependence are engaged in rowing (*n* = 6), team sports (*n* = 4), fitness (*n* = 2), running (*n* = 1), and playing tennis (*n* = 1), and that those from team sports (*n* = 25), fitness (*n* = 18), running (*n* = 9), rowing (*n* = 9), tennis (*n* = 4), and swimming (*n* = 3) [30] are symptomatic non-dependent. A study conducted by the Vardar revealed that a total of 11 athletes participated in branches of Judo, Taekwondo, and Karate, but exercise addiction prevalence has not been observed in these athletes [5].

A large majority of the athletes showed similar characteristics to symptomatic non-dependents and at risk of exercise dependence in the study: they were male, students, 18–22 years old, single, studying in high school, had 3–4 training days per week, did one training per day, had 60–90 min training times daily, were satisfied with their physical appearance, healthy, and successful. This could be explained with the specific dynamics and characteristics of fighting and defense sports, which require special discipline and regular work. The fact that there was no difference in all these variables also supported this idea.

Research on exercise dependence has several limitations since the terminology, definition, and evaluation scales are still not well developed. Further research is required to determine the factors that contribute to addiction, prevention, and treatment of addiction. It should be noted that the EDS was a screening tool to identify symptoms and not used to make any official diagnosis. The fact that the work was performed in the province of Elazig constituted a limitation in terms of research. Another limitation was that the sample group was only selected from three sports branches. In addition to these limitations, it may be beneficial to increase the generalizability of the data obtained by applying the EDS to a larger sample group when considering other areas of exercise and provinces.

## 5. Conclusions

In conclusion, it was estimated that this study would provide information about the prevalence of exercise dependence in different sports branches and make significant contributions to the literature in terms of revealing information about groups at risk for exercise addiction. It was also an important first step in understanding the differences in the symptoms of exercise dependence and the relationship between exercise dependence symptoms and sports branches. Finally, this study could be a guide to identify individuals who may be at risk of exercise dependence and thus may raise awareness in the implementation and guidance of lifelong prevention programs.

## Figures and Tables

**Table 1 sports-07-00052-t001:** Demographic characteristics of the participants.

Sample Characteristics	f	%	Sample Characteristics	f	%
**Gender**			**How many years of regular training do you have in your branch?**		
Male	87	59.6	1–2 years	50	35.5
Female	54	38.3	3–4 years	43	30.5
**Age**			5–6 years	28	19.9
18–22	84	59.6	7–8 years	5	3.5
23–27	29	20.6	9+ years	15	10.6
28–32	12	8.5	**How many days a week do you exercise?**		
33–37	8	5.7	Irregular	28	19.9
38+	8	5.7	1–2 days	24	17.0
**Marital Status**			3–4 days	61	43.3
Single	94	66.7	5–6 days	16	11.3
Married	29	20.6	7 days	12	8.5
Divorced	9	6.4	**How many times a day do you exercise?**		
Widow	9	6.4	1 training	84	59.6
**Education Level**			2 trainings	19	13.5
Primary school	14	9.9	3 trainings	26	18.4
Secondary school	13	9.2	4 trainings	6	4.3
High school	55	39.0	5+ trainings	6	4.3
Degree	45	31.9	**How long is your daily training time?**		
Graduate	14	9.9	Less than 30 m	16	11.3
**Sports Branch**			30–60 m	24	17.0
Kickboxing	76	53.9	60–0 m	51	36.2
Taekwondo	28	19.9	90–120 m	36	25.5
Muay Thai	37	26.2	120 m and over	14	9.9
**Job**			**Are you satisfied with your physical appearance?**		
Student	78	55.3	Yes	72	51.1
Official	18	12.8	Partially	54	38.3
Worker	14	9.9	No	15	10.6
Unemployed	23	16.3	**What is your the reason for training?**		
Self-employed	8	5.7	Like	13	9.2
**Sports Age**			To feel good	30	21.3
1–2 years	45	31.9	Being healthy	41	29.1
3–4 years	33	23.4	Succeed	47	33.3
5–6 years	28	19.9	Material gain	7	5.0
7–8 years	10	7.1	Other	3	2.1
9+ years	25	17.7			

**Table 2 sports-07-00052-t002:** Mean overall score of exercise dependence scale.

Scala Tool	*n*	Min	Max	Scala Score	Scala Value
Exercise Dependence Scale (EDS-21)	141	26.00	107.00	71.41	3.40

**Table 3 sports-07-00052-t003:** Comparison of mean overall score of exercise dependence symptoms (Kruskal–Wallis H test, one-way ANOVA).

Exercise Dependence Symptoms	*n*	%	$\bar{x}$	SD	F	*p*
Asymptomatic	5	3.5	34.00	0.40	150.882	
Symptomatic	117	83.0	69.00 ^a^	0.25		0.000 **
Dependent	19	13.5	96.11	0.32		

* The mean difference is significant at the 0.05 level. ** The mean difference is significant at the 0.01 level. ^a^ Significant difference with other groups.

**Table 4 sports-07-00052-t004:** Comparison of mean overall score of exercise dependence symptoms (Mann–Whitney U test and Kruskal–Wallis H test).

Sample Characteristics		Exercise Dependence Symptoms						Statistic			
		Asymptomatic		Symptomatic		Dependent		Total		U	*p*
		Count	%	Count	%	Count	%	Count	%	–	–
**Gender**	Male	5	5.7%	69	79.3%	13	14.9%	87	100.0%	–	–
	Female	0	0.0%	48	88.9%	6	11.1%	54	100.0%	–	–
Total		5	3.5%	117	83.0%	19	13.5%	141	100.0%	–	–
Mean Rank		44.00	–	72.92	–	66.26	–	–	–	2319	0.845
**Sample Characteristics**		–	–	–	–	–	–	–	–	**Chi-Square**	***p***
**Age**	18–22	1	1.2%	71	84.5%	12	14.3%	84	100.0%	–	–
	23–27	1	3.4%	23	79.3%	5	17.2%	29	100.0%	–	–
	28–32	1	8.3%	11	91.7%	0	.0%	12	100.0%	–	–
	33–37	1	12.5%	6	75.0%	1	12.5%	8	100.0%	–	–
	38+	1	12.5%	6	75.0%	1	12.5%	8	100.0%	–	–
Total		5	3.5%	117	83.0%	19	13.5%	141	100.0%	–	–
Mean Rank		105.60		70.18		66.95				4.906	0.086
**Marital Status**	Single	2	2.1%	77	81.9%	15	16.0%	94	100.0%	–	–
	Married	3	10.3%	22	75.9%	4	13.8%	29	100.0%	–	–
	Divorced	0	0.0%	9	100.0%	0	0.0%	9	100.0%	–	–
	Widow	0	0.0%	9	100.0%	0	0.0%	9	100.0%	–	–
Total		5	3.5%	117	83.0%	19	13.5%	141	100.0%	–	–
Mean Rank		84.40	–	72.14	–	60.45	–	–	–	2.732	0.255
**Education Level**	Prim. school	0	0.0%	14	100.0%	0	0.0%	14	100.0%	–	–
	Sec. school	0	0.0%	13	100.0%	0	0.0%	13	100.0%	–	–
	High school	1	1.8%	42	76.4%	12	21.8%	55	100.0%	–	–
	Degree	3	6.7%	35	77.8%	7	15.6%	45	100.0%	–	–
	Graduate	1	7.1%	13	92.9%	0	0.0%	14	100.0%	–	–
Total		5	3.5%	117	83.0%	19	13.5%	141	100.0%	–	–
Mean Rank		100.90	–	69.33	–	73.42	–	–	–	3.249	0.197
**Job**	Student	2	2.6%	64	82.1%	12	15.4%	78	100.0%	–	–
	Official	2	11.1%	15	83.3%	1	5.6%	18	100.0%	–	–
	Worker	1	7.1%	12	85.7%	1	7.1%	14	100.0%	–	–
	Unemployed	0	0.0%	21	91.3%	2	8.7%	23	100.0%	–	–
	Self-Employed	0	0.0%	5	62.5%	3	37.5%	8	100.0%	–	–
Total		5	3.5%	117	83.0%	19	13.5%	141	100.0%	–	–
Mean Rank		71.50	–	71.21	–	69.55	–	–	–	0.034	0.983
**Sports Branch**	Kickboxing	4	5.3%	64	84.2%	8	10.5%	76	100.0%	–	–
	Taekwondo	1	3.6%	17	60.7%	10	35.7%	28	100.0%	–	–
	Muay Thai	0	0.0%	36	97.3%	1	2.7%	37	100.0%	–	–
Total		5	3.5%	117	83.0%	19	13.5%	141	100.0%	–	–
Mean Rank		48.90	–	72.06	–	70.32	–	–	–	1.892	0.388
**Sports Age**	1–2 years	4	8.9%	37	82.2%	4	8.9%	45	100.0%	–	–
	3–4 years	1	3.0%	26	78.8%	6	18.2%	33	100.0%	–	–
	5–6 years	0	0.0%	22	78.6%	6	21.4%	28	100.0%	–	–
	7–8 years	0	0.0%	8	80.0%	2	20.0%	10	100.0%	–	–
	9+ years	0	0.0%	24	96.0%	1	4.0%	25	100.0%	–	–
Total		5	3.5%	117	83.0%	19	13.5%	141	100.0%	–	–
Mean Rank		30.80	–	72.53	–	72.16	–	–	–	5.337	0.069
**How many years have you been regularly training in your branch?**	1–2 years	5	10.0%	42	84.0%	3	6.0%	50	100.0%	–	–
	3–4 years	0	0.0%	32	74.4%	11	25.6%	43	100.0%	–	–
	5–6 years	0	0.0%	25	89.3%	3	10.7%	28	100.0%	–	–
	7–8 years	0	0.0%	4	80.0%	1	20.0%	5	100.0%	–	–
	9+ years	0	0.0%	14	93.3%	1	6.7%	15	100.0%	–	–
Total		5	3.5%	117	83.0%	19	13.5%	141	100.0%	–	–
Mean Rank		25.50	–	72.09	–	76.26	–	–	–	7.192	0.027 *
**How many days a week do you exercise?**	Irregular	3	10.7%	23	82.1%	2	7.1%	28	100.0%	–	–
	1–2 days	1	4.2%	21	87.5%	2	8.3%	24	100.0%	–	–
	3–4 days	1	1.6%	49	80.3%	11	18.0%	61	100.0%	–	–
	5–6 days	0	0.0%	13	81.3%	3	18.8%	16	100.0%	–	–
	7 days	0	0.0%	11	91.7%	1	8.3%	12	100.0%	–	–
Total		5	3.5%	117	83.0%	19	13.5%	141	100.0%	–	–
Mean Rank		33.40	–	71.12	–	80.16	–	–	–	5.743	0.057
**How many times a day do you exercise?**	1 training	5	6.0%	63	75.0%	16	19.0%	84	100.0%	–	–
	2 trainings	0	0.0%	16	84.2%	3	15.8%	19	100.0%	–	–
	3 trainings	0	0.0%	26	100.0%	0	.0%	26	100.0%	–	–
	4 trainings	0	0.0%	6	100.0%	0	0.0%	6	100.0%	–	–
	5+ trainings	0	0.0%	6	100.0%	0	0.0%	6	100.0%	–	–
Total		5	3.5%	117	83.0%	19	13.5%	141	100.0%	–	–
Mean Rank		42.50	–	75.53	–	50.63	–	–	–	11.023	0.004 **
**How long is your daily training time?**	<30 m	0	0.0%	16	100.0%	0	0.0%	16	100.0%	–	–
	30–60 m	1	4.2%	19	79.2%	4	16.7%	24	100.0%	–	–
	60–90 m	2	3.9%	40	78.4%	9	17.6%	51	100.0%	–	–
	90–120 m	2	5.6%	28	77.8%	6	16.7%	36	100.0%	–	–
	120+ m	0	0.0%	14	100.0%	0	0.0%	14	100.0%	–	–
Total		5	3.5%	117	83.0%	19	13.5%	141	100.0%	–	–
Mean Rank		75.90	–	70.65	–	71.84	–	–	–	0.095	0.954
**Are you satisfied with your physical appearance?**	Yes	1	1.4%	60	83.3%	11	15.3%	72	100.0%	–	–
	Partially	4	7.4%	42	77.8%	8	14.8%	54	100.0%	–	–
	No	0	0.0%	15	100.0%	0	0.0%	15	100.0%	–	–
Total		5	3.5%	117	83.0%	19	13.5%	141	100.0%	–	–
Mean Rank		86.90	–	71.62	–	63.03	–	–	–	1.863	0.394
**What is your the reason for training?**	Like	1	7.7%	9	69.2%	3	23.1%	13	100.0%	–	–
	To feel good	1	3.3%	28	93.3%	1	3.3%	30	100.0%	–	–
	Being healthy	3	7.3%	35	85.4%	3	7.3%	41	100.0%	–	–
	Succeed	0	0.0%	37	78.7%	10	21.3%	47	100.0%	–	–
	Material gain	0	0.0%	6	85.7%	1	14.3%	7	100.0%	–	–
	Other	0	0.0%	2	66.7%	1	33.3%	3	100.0%	–	–
Total		5	3.5%	117	83.0%	19	13.5%	141	100.0%	–	–
Mean Rank		45.50	–	69.97	–	84.03	–	–	–	4.262	0.119

* The mean difference is significant at the 0.05 level. ** The mean difference is significant at the 0.01 level.

**Table 5 sports-07-00052-t005:** Comparison of mean overall score of exercise dependence symptoms with variables differences (Kruskal–Wallis H, Post Hoc Test).

Dependent Variable	Exercise Dependence Symptoms (I)	Exercise Dependence Symptoms (J)	Mean Difference (I–J)	Std. Error	Sig.
**How many years have you been regular training in your branch?**	Asymptomatic	Symptomatic	–1.28205(**)	0.12130	0.000
		Dependent	–1.26316(**)	0.22739	0.000
	Symptomatic	Asymptomatic	1.28205(**)	0.12130	0.000
		Dependent	0.01889	0.25772	1.000
	Dependent	Asymptomatic	1.26316(**)	0.22739	0.000
		Symptomatic	–0.01889	0.25772	1.000
**How many times a day do you exercise?**	Asymptomatic	Symptomatic	–0.94017(**)	0.11079	0.000
		Dependent	–0.15789	0.08595	0.221
	Symptomatic	Asymptomatic	0.94017(**)	0.11079	0.000
		Dependent	0.78228(**)	0.14022	0.000
	Dependent	Asymptomatic	0.15789	0.08595	0.221
		Symptomatic	–0.78228(**)	0.14022	0.000

** The mean difference is significant at the 0.01 level.

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
