# Peer review of "Investigation of the Exercise Dependence of Athletes Doing Kickboxing, Taekwondo, and Muay Thai"

_sports, 2019, doi:10.3390/sports7020052_

Reviewer 1 Report

The authors present a descriptive research study examining exercise dependence / exercise addiction in martial arts athletes in Turkey. They report that ~14% of athletes demonstrate exercise dependence, while 83% of athletes have some symptoms of the condition. This is in line with some previous literature, although reports are highly variable.

Overall I found the manuscript somewhat hard to follow, especially in presentation of the methods and results. Specific comments are as follows, in the order they appear in the manuscript:

Introduction

(1) No hypothesis is given. While this is a descriptive study, the authors should report their pre-study expectations, whatever those were.

Methods

(2) Lines 100-102 are unnecessary/redundant

(3) The reasons for which the questionnaires were returned/withdrawn (lines 116-117) are not adequately explained

(4) The distinction between face-to-face interviewing and mail collection of responses (lines 121-122) are not adequately described. What data were collected face-to-face? What data were collected via mail? If some subjects were face-to-face and others via mail, does this bias the study results?

(5) The explanation of the study instrument is not satisfactory. For example, line 135-136 does not make sense. The method in which athletes were placed into categories on each specific sub-dimension seems to be explained. However, the method in which athletes were placed into an overall category is not clear.

Results

(6) Restating table data is not helpful. Specific items of interest should be described in the text.

(7) Line 164, "62.4% are trained to be healthy OR to be successful..."

(8) The categories for Physical Appearance in Table 1 are not easy to interpret.

(9) How was cause of training determined in Table 1? It seems likely that many athletes would give multiple reasons for this.

(10) Table 4 is hard to interpret and could be integrated into Table 3.

Author Response

Reviewer 1

Reviewer 2 Report

Thursday, January 30, 2019

Hello, and thank you for the opportunity to review your manuscript. Not only is this paper of particular interest to me, but I sincerely believe that this paper and its topic increasingly will be found of significant interests to athletes - particularly those in martial arts - around the world. That said, please consider the following suggestions for another draft of your paper and which I am happy to review:

Some of the English grammar and sentence structure will need to be finessed. However, for the most part, the English is used quite well. 

A suggested change to the grammar and sentence structure would be not to use contractions in formal writing like that used for this manuscript. For instance, in Lines 266, 282, and others, the complete words should be used rather than contractions.

Likewise, instead of using a term like "proven" (Line 30), you may want to use terms like "confirm" and "confirmed," as research may occur to discredit what was thought to be "proven" at an earlier time. 

Regarding the key words, perhaps you would consider "exercise dependence" rather than "dependence"? Or, if that is not possible, then perhaps "exercise" also should be included as a key word. 

In Lines 38 and 39, try not to overuse the word "However."

You should rework the sentence structure for Lines 59 - 60 to read more like, "...In contrast, women tend to be more addicted to weight loss as compared to men." This helps with clarity and order of thought. 

Also, try not to use the term "have" in front of your verbs. This will make your writing more influential because a more active form of the verb is used. For instance, in Line 67, you may want to write: "Szabo and Griffiths (2007) indicated..." Then, in Line 70, you may want to write: "Higher prevalence rates were reported..." In Lines 77-78, you may want to write: "Blaydon and Linder (2002) found that...: and so on and so forth throughout your paper. 

In Line 83, you made a very important point about the existence of an unclear definition of dependence, and this helps to establish a foundation for the main premise of the paper. 

In Lines 84 - 87, be sure to define with "it" is because "it" may refer to a number of different things. This confuses the reader. 

In Lines 89 - 90, please spell out completely each of the words for the EDS-R survey and then go ahead and use the abbreviation of EDS-R throughout the remainder of your paper. 

Why is the need for this study important to Turkish populations of martial artists? Of course, this is significant to Turkey. However, the argument could be made that the premise of the paper is important to all groups of people around the world. Therefore, why is this study particularly relevant to Turkey?

In Line 93, you should change "Kickbox" to "Kickboxing."

Throughout your paper - and provided the journal agrees - you may want to spell out the number if it is ten or less and then use the numbers for 11 or more. 

In Line 107, you may want to rephrase this as "The general screening model is a descriptive research method used in this research study" because this adds clarity.

In Line 116, why were 33 questionnaires withdrawn?

In Line 118, you may want to rephrase as "...The surveys of 141 athletes...were included in this study."

In Line 121, instead of writing "was applied to athletes," instead you may want to change this to "was administered to athletes."

In Line 128, would it be possible to include a copy of the EDS-21? Perhaps this could be included as an appendix?  The reader may have a clear idea of the measurement tool if a visual picture is included. 

In Lines 150 - 151, is the statistical package SPSS? SAS? Another? If so, then you may want to spell out the words for each initial.

In Line 160, you may want to write "...59.6% of respondents are..."

In Lines 160 - 165, you may want to use a more orderly manner of sharing the percentages, etc. because otherwise the current order may seem disorganized and somewhat random. 

For Table 1, perhaps you could rearrange this to make headings for each section? This way, the information contained within each is more obvious and less difficult to find? Perhaps expanding the table slightly through more space will add clarity.

In Lines 171 - 172, you might want to explain Scala. Is this a programming package? Is that what you mean by "Scala"? Clarity is needed. 

In Line 178, perhaps you want to say "one-way ANOVA" rather than "ANOVA" so as to distinguish this from a two-way ANOVA?

In Lines 185 - 193, try to follow a logical, similar order as what was included in Lines 160 - 165. Try to make it more obvious as to why you are presenting the information in this particular order. 

In Lines 275 - 277, use caution with this statement unless you know with certainty that martial arts athletes actually DO participate in other types of exercise to build endurance and strength. 

In Lines 300 - 303, for sake of clarity, you may want to rephrase this to something like, "Of the 19 athletes who are at risk for exercise dependence, eight athletes do kickboxing..."

Overall, the paper is extremely informative and will speak to athletes across various types of martial arts. This type of research is up and coming and should be published.

Thank you, and all of the best!

Author Response

revised appendix

Round  2

Reviewer 1 Report

The authors have appropriately responded to my comments for the most part. My one remaining concern is in the description of the overall dependence category assignment. It is still unclear how athletes were assigned to an overall category of asymptomatic, symptomatic, or dependent. Was it on the basis of the average of scores on the EDS-21, or some other measure? The authors make mention of a score of 5-6 on 3 or more criteria is an assignment for dependence, but it is not clear how this relates to the other categories. The cutoffs for this assignment should be specified.

Author Response

Dear reviewer;

Hausenblas and Downs defined;

The Exercise Dependence Scale-21 operationalizes exercise dependence based on the Diagnostic and Statistical Manual of Mental Disorder-IV (DSM-IV) criteria for substance dependence (APA, 1994) and provides the following information: 

(1)  Mean overall score of exercise dependence symptoms.

(2)  Differentiates between:  

(a)  at-risk for exercise dependence 

(b)  nondependent-symptomatic, and  

(c)  nondependent-asymptomatic. 

(3) Specifies whether individuals have evidence of:   (Not included in this study)

(a)  physiological dependence (i.e., evidence of tolerance or withdrawal) or  

(b)  no physiological dependence (i.e., no evidence of tolerance or withdrawal).

The proposed scoring procedure for the Exercise Dependence Scale is computer based which allows for immediate and accurate scoring. The computer scoring of the Exercise Dependence Scale is based on the SPSS (Statistic Package for the Social Sciences). The items are entered into SPSS enables:

 1. Computing a total and subscale mean scores for Exercise Dependence Scale-21. A higher score indicates more exercise dependent symptoms.

 2.  Categorizing participants into either at-risk for exercise dependent, nondependentsymptomatic, or nondependent-asymptomatic groups. The categorization into one of the three groups is generated by a scoring manual that consists of flowchart decision rules, in which items or combinations or items determine if an individual would be classified in the dependent, symptomatic, or asymptomatic range on each of the 7 DSM criteria. Individuals who are classified into the dependent range on 3 or more of the DSM criteria are classified as exercise dependence. The dependent range is operationalized as indicating a score of 5 or 6 for that item. Individuals who scored in the 3 to 4 range are classified as symptomatic. These individuals may theoretically be considered at-risk for exercise dependence. Finally, individuals who score in the 1-2 range are classified as asymptomatic.

(Hausenblas, H.A.; Downs, D.S. How much is too much? The development and validation of the exercise dependence scale. Psychol. Health. 2002, 17 (4), 387–404. https://doi.org/10.1080/0887044022000004894)

Let's try to explain.

Hausenblas and Downs did not provide cutoffs for the EDS-21. Athletes were generally assigned to an asymptomatic, symptomatic or dependent category on the basis of the average score in the EDS-21. The EDS-21 is a Likert type (never-1 and always-6) measure consisting of 21 questions developed to determine exercise dependence.

The total score of EDS-21 is between minimum 26 and maximum 126 points. Individuals who scored in the 0 to 42 range are classified as asymptomatic, who scored in the 43 to 84 range are classified as symptomatic and who scored in the 85 to 126 range are classified as dependent.

A cut-off score of 85 or more identifies individuals considered at risk for exercise dependence.